

# Are dense networks of low-cost nodes better at monitoring air pollution? A case study in Staffordshire

Louise Bøge Frederickson[1, 2, 3], Ruta Sidaraviciute[3], Johan Albrecht Schmidt[3], Ole Hertel[5], and Matthew Stanley Johnson[3, 4]

[1]Department of Environmental Science, Aarhus University, Frederiksborgvej 399, DK-4000 Roskilde, Denmark
[2]Danish Big Data Centre for Environment and Health (BERTHA), Aarhus University, Frederiksborgvej 399, DK-4000 Roskilde, Denmark
[3]AirLabs Denmark, Nannasgade 28, DK-2200 Copenhagen N, Denmark
[4]Department of Chemistry, University of Copenhagen, Universitetsparken 5, DK-2100 Copenhagen Ø, Denmark
[5]Department of Ecoscience, Aarhus University, Frederiksborgvej 399, DK-4000 Roskilde, Denmark

**Correspondence:** Matthew Stanley Johnson (msj@chem.ku.dk)

**Abstract.** Air pollution exhibits hyper-local variation, especially near emissions sources. In addition to people's time-activity patterns, this variation is the most critical element determining exposure. Pollution exposure is time-activity and path-dependent with specific behaviors such as mode of commuting and time spent near a roadway or in a park playing a decisive role. Compared to conventional air pollution monitoring stations, nodes containing low-cost air pollution sensors can be deployed with very high density. Monitoring stations are often tasked with characterizing regional air pollution and are therefore placed away from local sources, leaving the additional burden of local emissions such as traffic uncharacterized. In this study, a network of 18 nodes using low-cost air pollution sensors was deployed in Newcastle-under-Lyme, Staffordshire, UK, in June 2020. Each node measured a range of species including nitrogen dioxide ($NO_2$), ozone ($O_3$) and particulate matter ($PM_{2.5}$ and $PM_{10}$); this study focuses on $NO_2$ and $PM_{2.5}$ over a one year period from August 1, 2020 to October 1, 2021. A simple and effective temperature, scale and offset correction was able to overcome data quality issues associated with temperature bias in the $NO_2$ readings. In its recent update, the World Health Organization dramatically reduced annual exposure limit values from 40 to 10 $\mu$g m$^{-3}$ for $NO_2$ and from 10 to 5 $\mu$g m$^{-3}$ for $PM_{2.5}$. We found the average annual mean $NO_2$ concentration for the network was 17.5 $\mu$g m$^{-3}$, and 8.1 $\mu$g m$^{-3}$ for $PM_{2.5}$. While in exceedance of the WHO guideline levels, these average concentrations do not exceed legally binding UK/EU standards. The network average $NO_2$ concentration was 12.5 $\mu$g m$^{-3}$ higher than values reported by a nearby regional air quality monitoring station, showing the critical importance of monitoring close to sources before pollution is diluted. We demonstrate how data from a low-cost air pollution sensor network can reveal insights into patterns of air pollution and help determine whether sources are local or non-local. With spectral analysis, we investigate the variation of the pollution levels and identify typical periodicities. Both $NO_2$ and $PM_{2.5}$ have contributions from high-frequency sources, however, the low-frequency sources are significantly different. Using spectral analysis, we determine that at least 54.3 $\pm$ 4.3 % of $NO_2$ is from local sources, whereas in contrast, only 37.9 $\pm$ 3.5 % of $PM_{2.5}$ is local.



## 1 Introduction

According to the World Health Organization (WHO), seven million premature deaths every year can be attributed to poor air quality (Lelieveld, 2015; WHO, 2021). In response to the adverse health effects caused by air pollution, the WHO developed Air Quality Guidelines (AQGs) for a set of key air pollutants, including nitrogen dioxide ($NO_2$) and particulate matter with an aerodynamic diameter $\leq 2.5\ \mu$m ($PM_{2.5}$) (WHO, 2021). Since WHO's 2015 recommendation, evidence has accumulated showing many additional negative impacts of air pollution on health (Abdo, 2016; Sun, 2016; Chen, 2018; Ai, 2019; Wang, 2019; Wu, 2019; Zhang, 2020). After a comprehensive review of the evidence the WHO has recently recommended a much more strict set of standards and warned that exceeding the new air quality guideline levels is associated with significant health risks. **Table 1** shows the previous and revised AQGs for the pollutants of focus within this study along with the EU standards. These standards are legally binding, while the WHO values are indicative.

| Pollutant | EU | | AQGs 2015 | | AQGs 2021 | |
|---|---|---|---|---|---|---|
| | Averaging period | Concentration | Averaging period | Concentration | Averaging period | Concentration |
| $NO_2$ | Annual mean | 40 | Annual mean | 40 | Annual mean | 10 |
| | 1-hour mean | 200 | 1-hour mean | 200 | 24-hour mean | 25 |
| $PM_{2.5}$ | Annual mean | 25 | Annual mean | 10 | Annual mean | 5 |
| | | | 24-hour mean | 25 | 24-hour mean | 15 |

**Table 1.** Air quality standards set by the European Union (Gemmer, 2013) and WHO's global air quality guidelines (AQGs) from 2015 and 2021 (WHO, 2021). All concentrations are in $\mu$g m$^{-3}$.

Traditionally, air quality monitoring is based on static air quality monitoring stations (AQMS) with calibrated high-precision instruments. However, due to their purchase and maintenance costs, conventional AQMSs are generally sparsely located (Kumar, 2015; Maag, 2018). This monitoring strategy is suited to characterizing regional air quality but could fail to account for elevated concentrations near sources. Moreover the temporal and spatial resolution of such monitoring station networks is limited (Motlagh, 2020). For example there are a total of 18 AQMSs in the nation of Denmark, responsible for measuring concentrations at street level, urban background and regions (Danish National Monitoring Program for Water and Nature (NOVANA) (Ellermann, 2018)).

Meanwhile, field studies have shown that pollution levels, especially in urban environments, can vary substantially within a few meters due to localized air pollution sources (Lebret, 2000; Kingham, 2000; Monn, 2001; Zou, 2009; Wang, 2018; Li, 2019; Wilson, 2019). The local component can often be an important factor contributing to people's exposure, for example, for those who commute in a vehicle and/or work as professional drivers, street police, bicycle delivery etc. (Frederickson, 2020a), or live or work in buildings near busy roads. Low-cost air pollution sensors and sensor networks have evolved rapidly during the last few decades, enabled by technological progress and the development of fast and inexpensive wireless communication systems (Snyder, 2013). While the technologies are still evolving, low-cost air pollution sensors are becoming available and are starting to become a valuable supplement to the sparse conventional AQMS. Low-cost sensor (LCS) based networks are





not a substitute for networks of conventional AQMSs, since high-quality monitoring data is necessary for checking compliance with guidelines and they are also necessary for validating less expensive mapping obtained from modelling and/or LCS based monitoring.

Networks of low-cost air pollution sensors are becoming more common. On a device level, clearly the sensor elements cannot
compete with commercial instruments regarding The Three 'S's: Sensitivity, Stability, and Selectivity (Lewis, 2016; Borrego, 2016; Castell, 2017; Frederickson, 2020b), this may be more than compensated because LCSs enable greatly increased site density and temporal resolution, facilitating new insights into patterns and sources of air pollution. In addition, LCSs can supplement not only coarse-scale monitoring networks but also add substantial value to mappings provided by mathematical models. Dense networks of LCSs can be used for source apportionment and to distinguish local from non-local pollution
(Heimann, 2015), and as an aid in interpreting mathematical models that are often an integrated part of air quality monitoring (Hertel, 2007).

Within this study, electrochemical and semiconductor LCSs are used to measure gaseous pollutants and laser based particle counters are used to quantify particulate matter. Electrochemical and semiconductor sensor technology offer a number of advantages including linear response, small size, low cost in fabrication, relatively fast response, and low power consumption
(Frederickson, 2020b). While low-cost air pollution sensors bring new opportunities for monitoring, important issues remain regarding data quality. Studies show that sensor data can be influenced by environmental factors such as temperature and confounding gases (Spinelle, 2015, 2017; Mead, 2013; Bulot, 2020). Considerable efforts have been made to understand these factors, with varying success. Field work presents a complex and dynamic environment, greatly complicating the task of calibration. Experience shows that it is crucial to test each individual sensor and correct for multiple ambient factors (Popoola,
65 2016).

While a time series analysis based on summary statistics is a simple and effective tool, more sophisticated techniques are necessary to better understand the ultimate causes of these variations (Hwang, 2000). Spectral analysis using the Fourier transform can provide a deeper understanding of time series, because transformation into the frequency domain allows characterization of sources according to their periodicity and rate of change (Percival, 1998). While spectral analysis has long been used for
meteorological variables, because of its ability to distinguish synoptic and seasonal signals (Van der Hoven, 1957; Lyons, 1975; Eskridge, 1997), studies applying the Fourier transform to air pollution data emerged much later (Rao, 1976; Hogrefe, 2006; Choi, 2008; Lazi, 2016).

There is a relation between temporal and spatial scales of air pollution (Brasseur, 2017). Analysis of air quality data in the frequency domain contributes to the understanding of periodic behaviors and yields information about spatial and temporal
scales of the hidden, underlying mechanisms (Hies, 2000; Sebald, 2000; Marr, 2002). Short-term fluctuations of the pollutant concentrations are related to local-scale phenomena, including local dispersion conditions and patterns in local emissions and chemistry. Conversely, seasonal changes and the long-range transport and emissions of pollutants contribute to the spectrum at very low frequencies (Tchepel, 2009). On the time-scale of days, there are the motions of weather systems for example a high pressure system with well-developed photochemical air pollution. Pollution arriving from a distant source is characterized
by a slowly rising and falling signal due to the effects of transport time and atmospheric mixing. Regional emissions are of



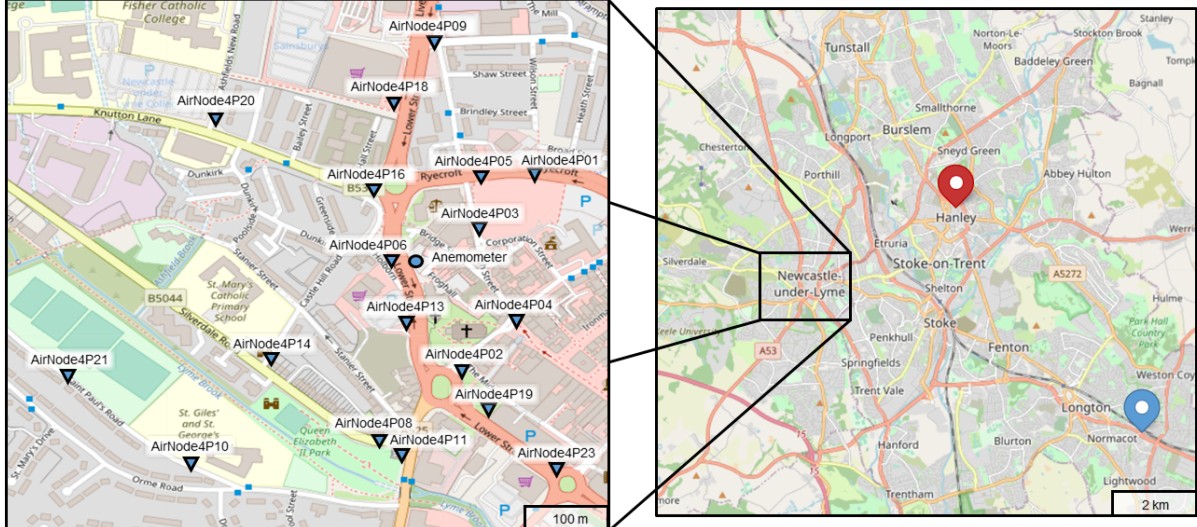

**Figure 1.** Spatial distribution of the AirNode network (left) and an overview of the location of the network relative to the two closest reference stations (right). The urban background station at Stoke-on-Trent Center is highlighted with a red marker, whereas roadside monitoring station at Stoke-on-Trent A50 Roadside is highlighted with a blue marker. The last AQMS used in this study (regional background monitoring station at Ladybower) is located 54 km from the network and is for clarity not included on the map. Maps obtained from ©OpenStreetMap contributors 2021. Distributed under the Open Data Commons Open Database License (ODbL) v1.0 (OpenStreetMap, 2021).

course regional in scale and photochemical pollution typically develops in a synoptic air mass. In contrast local sources (e.g. traffic) more often present as a sharp spike in concentration. Even an instantaneous puff of pollution will broaden with time based on the vertical and horizontal eddy diffusion coefficients $K$ which are on the order of 100 m$^2$ s$^{-1}$ (Seinfeld and Pandis, 2016). Using the Einstein-Smoluchowski relation $K = d^2/(2t)$ (Einstein, 1905; Smoluchowski, 1906), we can solve for the

characteristic distance as a function of time, $d = (2Kt)^{1/2}$. At a wind speed of 5 m s$^{-1}$, after a day, a spike of pollution will take a minimum of 15 minutes to pass.

  In this paper, we show how low-cost air pollution sensors provide additional insights into the patterns and sources of air pollution when deployed as a network rather than as individual sensors. A low-cost air pollution sensor network consisting of 18 low-cost air pollution sensor nodes (called *AirNode4PX*) was deployed in Newcastle-under-Lyme, UK, in the area centered

around the ring road (see **Figure 1**). The variation in road width, the different types of road structure, and highly variable traffic patterns all impact pollutant dispersion, resulting in significant spatiotemporal variation of pollution in the area. Each AirNode measured a range of species including nitrogen dioxide (NO$_2$), ozone (O$_3$) and particulate matter (PM$_{2.5}$ and PM$_{10}$); in this paper we focus on NO$_2$ and PM$_{2.5}$. The data obtained from the low-cost air pollution sensor network is used for time series analysis in the frequency domain to obtain information on the variability of air pollution concentrations and to distinguish local

sources from regional. The network, together with the analysis approach, has allowed pollutant emissions attributable solely to the local sources to be distinguished from other regional or long-range transport sources.



## 2   Field trial of the Staffordshire network

In June 2020, a network of 18 air pollution sensor nodes containing low-cost electrochemical and metal oxide gas sensors and
optical particle counters was deployed in Newcastle-under-Lyme in Staffordshire, UK, in the area centered around the ring
road. In addition, an anemometer was installed to record wind speed and direction. The initial 14 day installation, stabilization
and testing period of the measurement campaign are excluded from the analysis. In all the study covers a 14-month period
from August 1, 2020 to October 1, 2021.

### 2.1   Nodes of low-cost air pollution sensors

The nodes include low-cost air pollution sensors, signal processing and communications. The units, $88 \times 88 \times 90$ mm, are
assembled by AirLabs into weatherproof enclosures with full exposure to ambient air, and are set up to report measurements
to a cloud hosted by Amazon Web Services. The low-cost air pollution sensor nodes are generation 4P and are referred to as
*AirNode*, *AirNode4PX* or *4PX*, with $X$ being the node number. Each AirNode includes sensors for measuring $NO_2$ (NO2-
B43F from Alphasense Ltd.) and $O_3$ (MiCS-6814 from SGX Sensortech) as well as $PM_{2.5}$ and $PM_{10}$ (SDS-011 from Nova
Fitness Co.) at a 1-min time-resolution. In addition, each node is equipped with a control board and micro-controller unit
(ESP32) for programming the sensors. The AirNodes were mounted 2.5 to 3 m above street level on lamp posts which also
provided power as shown in **Figure 1**.

The SDS-011 sensor (Nova Fitness Co. Ltd, 2015) is a low-cost air pollution sensor measuring $PM_{2.5}$ and $PM_{10}$. Its principle
of operation is based on light scattering (van de Hulst, 1981), where particle density distribution is determined using the
intensity distribution patterns produced when particles scatter a laser beam (Liu, 2019). The sensor module includes a fan to
ensure a continuous flow of air through the sensor chamber (Genikomsakis, 2018). An algorithm converts the particle density
distribution into particle mass, and it can measure the particle density distribution between 0.3 to 10 $\mu$m (Bulot, 2020; Budde,
2018).

For $NO_2$ measurements, the NO2-B43F sensor (Alphasense, 2019) is used. This is an amperometric electrochemical gas
sensor containing four electrodes, where the principle of operation is based on electrochemistry (Frederickson, 2020b). When
the Working Electrode (WE) is exposed to ambient air, the target gas can diffuse onto the surface of the electrode, where it is
chemically reduced, resulting in a change in current. The Counter Electrode balances the current, and the Reference Electrode
sets the operating potential of WE. The fourth electrode is an Auxiliary Electrode (AE) and has the same structure as WE but
is not exposed to ambient air, hence is not affected by the target gas concentration, only by environmental parameters such as
temperature. Therefore, the difference in voltage between the WE and AE corresponds to changes in target gas concentration at
the EC cell surface. A trans-impedance amplifier converts the currents from the EC cell into a voltage. The voltage is amplified
further by a non-inverting operational amplifier, then a 16-bit analogue to digital (A/D) converter (ADS1115) samples the
output and produces a digital reading of the voltage level. This is used by the microprocessor to calculate the actual gas
concentration (Cross, 2017; Stetter, 2008; Mead, 2013). To minimize possible cross-interference from ozone, the $NO_2$ sensors
were fitted with integrated catalytic ozone filters ($MnO_2$ filters). The performance of these filters was verified in the laboratory,





and the $NO_2$ sensors showed no significant response to ozone in the range of $0 - 100$ ppb. Cross-interferences from other common gas pollutants were not considered important based on prior studies (Sun, 2017; Mead, 2013).

The MiCS-6814 metal-oxide sensor from SGX Sensortech (SGX Sensortech, 2015) includes three sensor chips with independent sensing layers and heaters. The three sensor chips are presented as the *NH3*, *RED* and *OX* sensor. The names of the chips relate to the type of gases the respective sensors are most suitable for detecting, i.e., $NH_3$ (NH3), reducing (RED) or

oxidizing (OX) gases, respectively. In the AirNodes, only the last two sensor chips are active. Chip OX and RED both consist of $WO_3$, whereas RED is also surface doped with Pd $(0.5 - 1 \%)$ leading to a higher sensitivity towards reducing gases SGX Sensortech (2015). The sensing mechanism is dependent on surface reactions. Therefore, the grain size of the sensor chip surface and the surface-to-volume ratio of the layer will alter the sensitivity of the MOx sensor. The thickness and porosity of the layer will also affect sensor response and recovery time from pollutant exposure. These effects are explained further in

other studies (Dey, 2018; Korotcenkov, 2008; Fine, 2010).

The temperature of the two active resistors on each sensor chip uses pulsed modulation. In this technique, two different voltages (1.8 V and 2.3 V) are applied to the resistor for 30 seconds each in an alternating way. A complete temperature cycle lasts one minute. The relationship between the actual sensor temperature and the values for heating power (76.7 mW and 45.6 mW) has been measured by SGX Sensortech. They estimate temperature to be approximately 360 °C in the high and 240

°C in the low-temperature period. Two sensor chips measure the sampled air in series, with a proprietary integrated $O_3$ filter between the two. This removes $O_3$ and therefore sensor chip 1 is exposed to both gases, whereas sensor chip 2 (after the filter) is exposed to $NO_2$ without the presence of $O_3$ (Viricelle, 2006). The output of both sensor chips is then used in determining the concentrations of the gases, and the cross-sensitivity can be mitigated. Active sampling is crucial for sensor performance since the sensing elements consume the target gas when measuring, therefore if the surrounding air is not moving, a lower and

possible non-linear response would be observed.

## 2.2 Correction methodology

The calibration of the electrochemical sensors measuring $NO_2$ is known to vary at high $(>20°C)$ and low $(< 0°C)$ temperatures and with rapid temperature change (Alphasense, 2019; Popoola, 2016; Li, 2021). Therefore we apply a correction with coefficients determined by using a linear regression model:

$$NO_2 \, (cor_T) = a_0 + a_1 \cdot T + a_2 \cdot dT/dt + a_3 \cdot NO_2 \, (raw) \tag{1}$$

where $NO_2$ (raw) are the raw $NO_2$ readings obtained from the AirNodes. $T$ is filtered temperature data obtained from the nearest reference station. Filtered temperature represents the temperature reading when ambient temperature exceeds $10°C$ and is transformed according to

$$f(t) = \begin{cases} 0 & \text{if } T < 10 \\ T - 10 & \text{otherwise} \end{cases} \tag{2}$$



and $dT/dt$ is the rate of change of the filtered temperature. The temperature threshold of 10°C was chosen because the internal temperatures of the LCS nodes often exceed the ambient temperatures and the performance of the correction was sufficient. The linear regression coefficients or correction coefficients, $a_0$, $a_1$, $a_2$ and $a_3$, are calculated using the method of multiple least squares, separately for each AirNode (Spinelle, 2017).

All electrochemical sensors have a different inherent sensitivity, hence the $NO_2$ readings need to be scale-corrected. The
scale-correction is carried out by multiplying the temperature-corrected $NO_2$ readings ($NO_2$ ($cor_T$)), from each AirNode, with $\alpha$, which is the ratio between the 0.80 and 0.20 quantiles of the $NO_2$ readings obtained from the AirNodes ($Q_{\text{diff, AirNode}}$) and from the reference ($Q_{\text{diff, Reference}}$). The reference is the $NO_2$ readings obtained by chemiluminescence from the reference-grade instrument at the AQMS at Stoke-on-Trent Centre, 4.1 km from the network, from the same period as the measurements took place. The difference between the 0.80 and 0.20 quantiles is a proxy for the variation obtained in the measurements.

$$Q_{\text{diff, AirNode}} = Q_{0.80, \text{ AirNode}} - Q_{0.20, \text{ AirNode}} \tag{3}$$

$$Q_{\text{diff, Reference}} = Q_{0.80, \text{ Reference}} - Q_{0.20, \text{ Reference}} \tag{4}$$

$$\alpha = Q_{\text{diff, Reference}} / Q_{\text{diff, AirNode}} \tag{5}$$

$$NO_2 \ (cor_{T,S}) = NO_2 \ (cor_T) \cdot \alpha \tag{6}$$

The offsets of the readings are determined by calculating the difference between the 0.25 quantile ($Q_{0.25}$) obtained from
each AirNode ($Q_{0.25, \text{AirNode}}$) and from the reference ($Q_{0.25, \text{ Reference}}$). Hence, the offset of the temperature- and scale-corrected reading ($NO_2$ ($cor_{T,S}$)) is adjusted by subtracting the calculated offset ($\beta$). The reference used in the offset-correction is the same as the one used for the scale-correction. The 0.25 quantile is a proxy for measured background concentration.

$$\beta = Q_{0.25 \text{ AirNode}} - Q_{0.25, \text{ Reference}} \tag{7}$$

$$NO_2 \ (cor) = NO_2 \ (cor_{T,S}) - \beta \tag{8}$$

where $NO_2$ ($cor_{T,S}$) is the temperature- and scale-corrected $NO_2$ reading, and $NO_2$ (cor) is the temperature-, offset- and scale-corrected $NO_2$ reading.

### 2.3 Comparison with regulatory air quality monitoring stations

The data obtained from the network is compared with data from the three nearest regulatory air quality monitoring stations: The roadside monitoring station at Stoke-on-Trent A50 Roadside (52.980436°N, 2.111898°W, 8.7 km from the network), the
urban background monitoring station at Stoke-on-Trent Centre (53.028210°N, 2.175133°W, 4.1 km from the network) and the regional background monitoring station at Ladybower (53.403370°N, 1.752006°W, 54 km from the network). We do not expect perfect agreement but nonetheless the exercise is useful.

Ladybower is located in the Peak District National Park around 800 meters to the southwest of Ladybower reservoir. The nearest road is 20 meters from the station and is only used by the nearby farmsteads. The surrounding area is mainly open
moorland. The urban background monitoring station located in Stoke-on-Trent is in the northern part of downtown Hanley. This




station is located five meters from a road connected to a busy multi-story car parking facility (50 meters from the monitoring station). The surrounding area is open grass with a few trees and commercial properties. The A50 Potteries Way is a busy ring road which lies approximately 130 meters to the north-east of the monitoring site. The roadside monitoring station is located between the main road and a parallel side road, near a pedestrian footbridge, beside the dual carriageway A50 through Stoke.

All three AQMSs are equipped with instruments for measuring $NO_2$ by chemiluminescence, but only Stoke-on-Trent Centre measures $PM_{2.5}$. Hourly air pollution data from each monitoring station were manually downloaded using the UK-Air data selector (DEFRA, 2022).

## 2.4 Spectral analysis

Spectral analysis is widely used for investigating cycles and variations of pollutants in time series to reveal the sources of 
pollution (Marr, 2002; Lazi, 2016). Within spectral analysis, the Fourier transform is a powerful tool for analyzing time series including periodicities and rate of change. To use the method it is necessary to overcome obstacles including the often unevenly spaced time points in time series due to technical and practical problems during monitoring (Sun, 1996, 1997). The unequally spaced or missing data can be circumvented by applying the fast Fourier transform after filling the gaps and missing values with the mean. In addition, the linear trend in the time series is removed by subtracting the average concentration obtained by 
each LCS. The periodogram for a finite time series is calculated as the square of the magnitude of $X$

$$\Phi(\nu_k) = |X(k)|^2 = \left| \frac{1}{\sqrt{N}} \sum_{t=0}^{N-1} x_t e^{(-2\pi i \nu_k t)} \right|^2 \tag{9}$$

where $k = 0, 1, \cdots, (N-1)$, $N$ is the number of observations, $x_t$ is the segment time series, and $n = k/N$. The periodogram indicates the strength of the signal as a function of frequency, while its spectrum over the frequency range corresponds to the variance of the time series data. Parseval's Theorem (Parseval, 1806; Narayanan, 2003) states that the energy, or in this case 
intensity, is conserved during Fourier transformation. Thus, the contribution of the different pollution sources can be quantified by integrating the peaks in the periodograms (Marr, 2002).

There is a relationship between temporal and spatial scales of the different air pollutants. Rapid, short-term fluctuations of the pollutant concentrations happen as a result of local phenomena, e.g. local-scale dispersion, local emissions and short-term atmospheric chemistry. Rapid changes contribute to the periodogram at high frequencies, which, in this work are defined as 
above $0.0417 \, \text{h}^{-1}$, i.e., events with a frequency higher/shorter than one day. This is referred to as the 'local' contribution to the pollutant concentration. The seasonal changes in the emissions and long-range transport of the pollution contribute to the periodogram at low frequencies ($< 0.0139 \, \text{h}^{-1}$), i.e., events with a frequency lower/longer than three days. This is then referred to as the 'regional' contribution to the pollutant concentration. In this model, intermediate frequencies are due to the 'urban' contribution to the pollutant concentration. As noted in the introduction, the mixing of pollution with time provides a upper 
limit on frequency for distant sources; only local sources can give a high frequency signal. The above-mentioned definitions are illustrated on **Figure 2**. One of the properties of diffusion is that a pulse of pollution will propagate in a Gaussian concentration profile depending on the diffusion constant and time. Under the Fourier transform, a Gaussian is mapped onto another Gaussian





with a different width. The transform of a wide function is narrow and vice versa. By integrating the peaks in the three different frequency bins, the relative contribution of local, urban and regional pollution of the LCS data can be quantified. After the
relative contributions are calculated for each LCS node, the average concentration together with the standard deviation can be calculated across the AirNode network to illustrate how much local pollution the network is seeing on average and how much variation is seen across all AirNodes.

## 3    Results and discussion

In the following section we present the results of our study and of the data analysis.

### 3.1    Sensor data quality

The first requirement is to establish the fidelity of the monitoring network.

#### 3.1.1    Missing data

The data completeness of the AirNodes varies between sites. In the monitoring network, apart from four AirNodes (4P04, 4P06, 4P08 and 4P20), all AirNodes have more than 80% data completeness during the sampling period. The four AirNodes with
data completeness below 80% were excluded from the analysis. Across the network of AirNodes, the mean data completeness is 95%, which is sufficient to investigate the local variation of air pollution. The main reasons for data gaps are the irregularities in the line power and lapses in the wireless internet connection. In addition, spiders had in a few cases entered through the small holes at the base of the AirNodes and nested in the housing leading to sensor failure.

#### 3.1.2    Correction of $NO_2$ readings

It is necessary to account for temperature bias while deploying electrochemical $NO_2$ sensors (Alphasense, 2014). For our study, this correction was crucial in order to get meaningful readings from the electrochemical sensors since the raw readings showed unphysical behavior. The typical $NO_2$ patterns during weekdays (Monday to Friday) and weekends (Saturday and Sunday) measured by AirNode4P01 as an example are shown before and after the correction in **Figure 3**. The $NO_2$ patterns of the corrected $NO_2$ readings compared to the readings from the reference indicate that the correction procedure can overcome most
of the disparity between the readings during higher temperatures. Modeled temperature data from DEFRA (DEFRA, 2022) are used to correct for the temperature bias. The correction coefficients for the AirNodes were calculated for each individual AirNode and the mean and standard deviation of the correction coefficients are: $a_0$ = 20.83 (13.29), $a_1$ = -0.30 (0.17), $a_2$ = 1.37 (0.58), $a_3$ = 892.38 (521.02). The relatively high standard deviations are linked to the known intra-sensor variability and show that each sensor requires individual calibration. It is known that in cities the temperature can vary strongly over small
distances (Cao, 2021), therefore it would have been more accurate to measure the internal temperature of the AirNodes and use that information for the correction. However, the correction methodology even with the modeled temperature data, yields corrected readings that follow expected trends, giving confidence in sensor accuracy. However, as seen in **Figure 3**, there is



still a relatively large discrepancy between the reference and the corrected AirNode readings on weekdays between 8 and 12 h, which can be attributed to the large distance between the reference instrument and the AirNodes (4.1 km) and the fact that

the concentration of $NO_2$ can have different profiles at different locations, depending on the traffic modes and sources. Sensor performance is validated below.

### 3.1.3 Inter-sensor variability

Inter-sensor variability has been used as a metric of sensor reliability in recent studies (Liu, 2020). The Pearson correlation coefficients for $PM_{2.5}$ among the AirNodes ranged from 0.87 to 0.99 with a mean of 0.95. In contrast the Pearson correlation

coefficients for $NO_2$ ranged from 0.30 to 0.88 with a mean of 0.64. For $PM_{2.5}$, the lowest Pearson correlation coefficients were above our quality criteria of 0.85. We did not choose a similar criterion for $NO_2$ since we expect there is much higher variation between the sensors due to localized sources. The AirNode network readings rose and fell simultaneously as ambient concentrations and conditions changed confirming that the sensors are operating as expected and giving confidence in sensor measurement reliability. In addition, this indicates that the AirNodes meet the specifications of the Class 1 device standard

specifying quality objectives for indicative measurements (AQ, 2021). Class 1 dictates that measurement uncertainty should be below 25% for $NO_2$ and 50% for $PM_{2.5}$. 'Indicative measurement' refers to the definition in Directive 2008/50/EC (EU, 2008).

### 3.2 Descriptive statistics

Air quality data for $NO_2$ and $PM_{2.5}$ measured at the different sites during 2020 and 2021 were analyzed. For this section, only

one year of data (August 1, 2020 to August 1, 2021) is used to compare with official guidelines. Descriptive statistics of the air quality measurements are presented in **Table 2**. The mean concentrations are compared to WHO's recently updated European AQGs and the legally binding EU standards, see **Table 1**. It should be noted that the legally binding values for annual means are defined for January, 1 to December, 31. The mean annual $NO_2$ and $PM_{2.5}$ concentrations across the network exceed the updated WHO guidelines by 7 and 3 $\mu$g m$^{-3}$ for $NO_2$ and $PM_{2.5}$, respectively. All sites have days where the daily average

$NO_2$ and $PM_{2.5}$ concentrations exceed the WHO daily average AQG limits during the period. However, none of the sites are above the legally binding EU standards.

The values obtained from the network can be analyzed in relation to the values reported by AQMSs, as long as the significant distance between the measurement locations is kept in mind. The values from the network are compared with the three AQMSs: the monitoring station at Stoke-on-Trent A50 Roadside, the urban background monitoring station at Stoke-on-Trent Centre

and the regional background monitoring station at Ladybower. Only the urban background station at Stoke-on-Trent Centre, reported $PM_{2.5}$. Concentrations of $NO_2$ and $PM_{2.5}$ (when available) are averaged within the same period as the AirNodes in the network (i.e. August 1, 2020 to August 1, 2021), and the descriptive statistics are shown in **Table 2**. The mean concentrations obtained from the network show similar values for $NO_2$ and $PM_{2.5}$ as those seen at the urban background station. On average while the network sees lower $NO_2$ values than the roadside monitoring station (21.4 $\mu$g m$^{-3}$), there is an $NO_2$ excess relative

to the regional background exposure (12.5 $\mu$g m$^{-3}$). In an environment such as a city with an elevated urban background





| Sensor | NO$_2$ concentration ($\mu$g m$^{-3}$) | | | | | PM$_{2.5}$ concentration ($\mu$g m$^{-3}$) | | | | |
|---|---|---|---|---|---|---|---|---|---|---|
| | Mean | Median | SD | Max | Exceedance[a] | Mean | Median | SD | Max | Exceedance[b] |
| AirNode4P01 | 18 | 16 | 11 | 82 | 41 | 9 | 5 | 12 | 122 | 58 |
| AirNode4P02 | 17 | 11 | 12 | 101 | 45 | 7 | 4 | 8 | 101 | 26 |
| AirNode4P03 | 18 | 15 | 11 | 89 | 44 | 9 | 6 | 10 | 121 | 40 |
| AirNode4P05 | 17 | 16 | 10 | 78 | 37 | 8 | 4 | 10 | 109 | 42 |
| AirNode4P09 | 19 | 16 | 13 | 91 | 51 | 8 | 4 | 11 | 119 | 44 |
| AirNode4P10 | 19 | 18 | 10 | 100 | 36 | 10 | 5 | 15 | 241 | 54 |
| AirNode4P11 | 18 | 17 | 10 | 71 | 36 | 11 | 6 | 13 | 125 | 43 |
| AirNode4P13 | 17 | 11 | 13 | 100 | 34 | 7 | 4 | 9 | 94 | 27 |
| AirNode4P14 | 17 | 10 | 13 | 91 | 60 | 7 | 4 | 9 | 92 | 19 |
| AirNode4P16 | 15 | 10 | 11 | 89 | 34 | 8 | 5 | 10 | 132 | 27 |
| AirNode4P18 | 17 | 14 | 11 | 86 | 43 | 7 | 4 | 11 | 163 | 33 |
| AirNode4P19 | 18 | 16 | 11 | 95 | 38 | 6 | 3 | 8 | 89 | 33 |
| AirNode4P21 | 18 | 15 | 10 | 83 | 47 | 7 | 4 | 11 | 140 | 32 |
| AirNode4P23 | 18 | 16 | 10 | 77 | 30 | 9 | 5 | 13 | 129 | 56 |
| Network mean | 17.5 | 14.4 | 11.2 | 88.0 | 41.1 | 8.1 | 4.3 | 10.7 | 126.8 | 38.1 |
| Network SD | 0.9 | 2.5 | 1.0 | 9.1 | 7.9 | 1.4 | 0.9 | 1.9 | 38.6 | 12.1 |
| Regional station | 5 | 4 | 4 | 45 | 0 | - | - | - | - | - |
| Urban station | 19 | 16 | 13 | 102 | 64 | 8 | 6 | 7 | 74 | 6 |
| Roadside station | 39 | 35 | 25 | 155 | 271 | - | - | - | - | - |

[a] Number of days with an average NO$_2$ concentration above WHO's guideline of 25 $\mu$g m$^{-3}$

[b] Number of days with an average PM$_{2.5}$ concentration above WHO's guideline of 15 $\mu$g m$^{-3}$

**Table 2.** Statistics for air quality data measured from Aug 01, 2020 to Aug 01, 2021. For comparison, the descriptive statistics from the three regulatory air quality monitoring stations (Regional = Ladybower, urban background = Stoke-on-Trent Centre, roadside = Stoke-on-Trent A50 Roadside) are shown for the corresponding period. Neither Ladybower nor Stoke-on-Trent A50 Roadside has instruments for monitoring PM$_{2.5}$. Abbreviations: SD = standard deviation, max = maximum value. We are aware that the measurement uncertainty is significantly higher for low-cost air pollution sensors than for reference air quality monitoring measurements. However, EU air quality guidelines approve low-cost sensor data as indicative but not quantitative data - in line with calculations with air quality models.

concentration, exposure to air pollution in micro-environments can cause exceedance of recommended threshold values for many individuals in addition to the dangers of transient and continued exposure.

## 3.3 Temporal trends

**Figure 3** shows the temporal variation in NO$_2$. On weekdays, the NO$_2$ concentration increases in two time periods during the
day, with peaks at 7:00 and 18:00. On weekends, the NO$_2$ concentration rises steadily throughout the day. There is a notable





decrease in concentration during the weekend compared to the weekdays at all sites. For both weekends and weekdays the NO$_2$ concentration is lowest at night. The two time periods with increased NO$_2$ concentration during the weekdays are typically periods of increased traffic during morning and afternoon rush hours when people commute to and from work (Vignati, 1996; Berkowicz, 1996). Thus, traffic likely drives this observed variation, in line with the declining NO$_2$ concentration during the

night and over the weekend.

In terms of monthly trends, **Figure 5** displays the monthly average of the NO$_2$ concentration. The highest NO$_2$ concentrations are seen in the Spring and the Winter. To fight against the spread of COVID-19, the United Kingdom implemented lockdown strategies, which resulted in a reduction of transportation, industrial, and commercial activities (Fluharty, 2021). **Figure 5** displays the monthly readings from one of the AirNodes together with the monthly readings from the nearest urban

background AQMS (Stoke-on-Trent Center). The readings from the AirNode and the AQMS follow the same trends. The lower NO$_2$ levels in the middle of the year possibly were a result of the lockdowns (Venter, 2020). Note that we also see seasonal variation thus the drop in NO$_2$ concentration might not solely be due to the COVID-19 pandemic.

## 3.4    Spatial trends

Wind speed and direction have been shown to provide essential information that can help identify source location (Carslaw,

2006; Westmoreland, 2007). The description of variation with wind direction and wind speed on a specific street (the so-called *street canyon effect*) is described in Berkowicz (1996). Bivariate polar plots are a powerful tool for source characterization including mean pollutant concentrations for specific wind speed and direction bins (Uria-Tellaetxe, 2012; Grange, 2016; Carslaw, 2012, 2013). In these plots wind direction is displayed from 0 to 360°clockwise on the angular axis and wind speed is shown on the radial scale.

The wind speed and direction data used in this study are shown as a windrose in **Figure 6**. The windrose shows that the prevailing winds come from the south and northwest during the measurement period. To assess spatially-resolved source patterns, bivariate polar plots of the NO$_2$ and PM$_{2.5}$ are investigated. The bivariate polar plots for each pollutant for all sites are shown in **Figure 7**. Reddish colors represent higher values compared to the blueish ones.

The bivariate polar plots show patterns that depend on deployment location. AirNode4P23, AirNode4P19 and AirNode4P02

are located in the southern part of the ring road, and they display similar patterns in their bivariate polar plots. Their surroundings are almost identical and the traffic influence on their readings is similar. The nodes located in the northern part of the ring road have different patterns relative to the ones in the southern area. They experience the highest values at lower wind speeds. When peak concentrations occur at low wind speeds it suggests local sources. For example in a street canyon there is both a direct and a recycled contribution to the concentration, where the relative size of these two contributions depends on whether

the measuring site at a given time is in the leeward or windward side of the street. AirNode4P10 is located in front of a school, and at lower wind speeds or with westerly wind, elevated levels of NO$_2$ were observed. In general, the highest concentrations are observed at low wind speeds, where no whirlwind is formed inside the street, independent of wind directions, or when the measuring site is on the leeward side of the street (in relation to the whirlwind). In the latter case, pollution from the traffic



in the specific section of the street will be led directly to the measuring site, at the same time as there is a contribution due to

trapping of pollution within the limited volume of the whirlwind.

Higher $NO_2$ values are correlated with wind speed and the orientation of the road. The traffic comes from the ring road area and continues through St. Paul's Road. Near the school, traffic stops frequently and accelerates and idles while children are being dropped off and picked up. The lowest values of $NO_2$ are seen at higher wind speeds with northwesterly winds. The bivariate polar plot for AirNode4P01 and AirNode4P05 show similar patterns, with the highest concentrations found for

easterly and southwesterly winds, whereas the lowest concentrations were seen with westerly and southwesterly winds. Higher speed southwesterly winds contributed to the peak concentrations at these locations. A wide-open parking area is located next to the ring road in that direction, which could explain the elevated concentrations.

The wind speed dependence of concentrations in a street canyon can be complex as there are opposing effects: Higher wind speed lead to more $O_3$ but also more dilution of $NO_x$ (NO + $NO_2$). High wind speeds will therefore lead to lower $NO_2$ while

at low wind speeds, $NO_2$ formation is limited by $O_3$, which goes towards zero in the street (Palmgren, 1996). Bivariate polar plots are good at revealing these interrelationships. The wind speed dependence can help distinguish sources from one another. When several measurement sites are available, polar plots can triangulate different sources (Carslaw, 2006). As expected, $NO_2$ is dominated by local emissions, and peak values mainly occur for low wind speeds, where elevated concentrations were observed due to accumulation and lack of dispersion. The most obvious features of $NO_2$ bivariate polar plots are that the

elevated levels are attributable to the orientation of the road or the place with the highest traffic density.

Relative to the bivariate polar plots of $NO_2$, the bivariate polar plots of $PM_{2.5}$ do not show as much variation across the network. Generally, the highest concentrations of $PM_{2.5}$ are seen for southeasterly winds and higher wind speeds. This is confirmed by the frequency spectrum showing slow changes consistent with large air masses. This indicates that particles originate from long-range transport. The bivariate polar plots for $PM_{2.5}$ also suggest that the locally-sourced particulate matter

is present, shown by the elevated concentrations at low wind speeds, where the atmospheric conditions are more stable.

In general, sites across the sensor network show a variation in their bivariate polar plots (however more for $NO_2$ than for $PM_{2.5}$) due to the different pollution sources. Thus, there are additional benefits of multi-sensor node measurements for characterizing sources in detail, especially when combining them with meteorological information.

**Figure 8** shows data from the urban background AQMS at Stoke-on-Trent Centre at a time-resolution of 1 hour (DEFRA,

2022). Data from one of the AirNodes with a time-resolution of 30 minutes is shown for comparison. The raw readings from the AirNodes have a time-resolution of 1 minute, however, the temperature correction aggregates the data into 30 minute bins. Still, with a time-resolution of 30-minutes, we see more local variability in the data, compared to the readings from the reference station. The data have a measurement density in both time and space, which can not be achieved using current conventional measurement methods. As seen on **Figure 8**, the readings from the AirNode and the AQMS follow the same trend, but the

correlation of determination is only 0.28. This is expected since the AQMS is located around 4 km from the AirNode network. However, increasing the time-resolution will increase the correlation of determination, i.e., a time-resolution of 3 hours results in a correlation of determination of 0.38, and a time-resolution of 1 day yields 0.63.



## 3.5 Spectral analysis

Spectral analysis is performed on the air pollution data to investigate its hidden periodicities and quantify their magnitude. The
contributions of local and regional sources to the pollution concentrations are determined based on the determined amplitudes
and frequencies. The local sources are shown in the high-frequency periodogram, and the regional or long-range sources are
revealed in the low-frequency periodogram. Note however that local sources may be present in both the low and high frequency
regions. For example, in an urban street, the traffic patterns follow stable patterns with daily and weekly periodicities. Holiday
periods follow their own pattern, and for wood smoke, emissions follow variations in outdoor temperature. By comparing the
spectra for the different pollutants measured by the same AirNode, information on the sources can be revealed. If the emission
sources for the different pollutants are the same, similarly cyclic patterns can be expected. The differences in the pollution
spectra can indicate a contribution from the different pollutant sources or the presence of chemical transformation since all
other conditions are identical.

Spectral analysis is performed on $NO_2$ data from three different types of AQMSs to illustrate how periodograms vary
depending on location. These AQMSs are 1) regional background (Ladybower), 2) urban background (Stoke-on-Trent Centre)
and 3) street (Stoke-on-Trent A50 Roadside). The three periodograms are shown in **Figure 9**. While all three periodograms
have significant peaks in the low-frequency region, only the urban background and street AQMSs have significant peaks in
the high-frequency region. We conclude that these high frequency peaks are due to the proximity and strength of local $NO_2$
sources.

**Figure 10** displays the periodograms for $NO_2$ and $PM_{2.5}$ measured by AirNode4P01. The periodogram of $NO_2$ features
three distinct peaks at 0.125, 0.084 and 0.042 $h^{-1}$ corresponding to periods of 8, 12 and 24 h, respectively. In addition, one
peak is identifiable in the high-frequency region at 0.17 $h^{-1}$ (6 h). In the low-frequency region, there are multiple peaks close
to each other, however, the peaks corresponding to 5 days (0.0083 $h^{-1}$), 1 week (0.0061 $h^{-1}$) and 1 month (0.00135 $h^{-1}$)
can still be identified. All these cycles can be related to local sources of pollution e.g. traffic or meteorological changes. Peaks
located in the low-frequency region can originate from changes over either synoptic or larger scale. Highest intensity odccurs
in the high-frequency region since most of $NO_2$ originates from local sources. The daily changes in $NO_2$ concentrations can be
associated with the daily changes in traffic from nearby roads and the diurnal variation caused by sunlight. Weekly periodicity
may also originate from changes in traffic.

The periodogram for $PM_{2.5}$ (see **Figure 10**) features one distinct peak in the high-frequency region at 0.042 $h^{-1}$ (24 h),
and a prominent peak at 0.084 $h^{-1}$ (12 h). Besides these two peaks, most peaks are seen in the low-frequency region of
the periodogram, which is expected since $PM_{2.5}$ is dominated by long-range transport and non-local sources. However, the
contribution by PM from a nearby road can originate from traffic since vehicles, in general, can re-suspend particles from the
road into the air, and abrasion from brakes and tires also produce PM (Grigoratos, 2015).

Periodograms for the rest of the AirNodes in the network show results similar to the ones shown in **Figure 10**, with small
changes in position and amplitude at specific locations. Conclusions regarding trends in pollution sources can be drawn by
examining the relative contributions from local, urban and regional sources. **Figure 11** shows the calculated percentages of





local, urban and regional contributions for the AirNodes as well as for the three different types of AQMSs. The results for the network indicate that local emissions are the most important source of $NO_2$ with an average of $54.3 \pm 4.3$ %, whereas $PM_{2.5}$ is mainly due to regional sources ($62.1 \pm 3.5$ %). For $NO_2$, urban sources contribute $14.3 \pm 1.9$ % and regional sources $31.2 \pm 4.5$ %. For $PM_{2.5}$, urban sources contribute $20.0 \pm 1.2$ % and local sources $17.9 \pm 3.2$ %.

As expected the regional background AQMS shows the highest relative contribution from of regionally sourced $NO_2$, and the street AQMS has the highest level of locally sourced $NO_2$. The AirNodes in the network show a distribution of contributions.

The results obtained for both $NO_2$ and $PM_{2.5}$ reveal contributions of short-term (12 h and 24 h) and long-term fluctuations. The contributions at low frequencies are significantly different between the two pollutants, indicating that temporal variations are influenced by different processes. The methodology is a powerful tool for analyzing the causes of air pollution.

## 4 Conclusions

Air pollution can be hyper-local and low-cost air pollution sensors are capable of accurately describing variation close to pollution sources. This study assessed more than one year of $NO_2$ and $PM_{2.5}$ data with high spatiotemporal resolution (1-min) obtained using a network of 18 low-cost air pollution sensor nodes. Initially there were significant calibration issues associated with temperature bias in the $NO_2$ readings but a simple and effective temperature, scale and offset correction was able to overcome this problem.

In its recent update and revision of the air quality guidelines for Europe, the WHO has proposed annual $NO_2$ and $PM_{2.5}$ exposure guideline thresholds of 10 and 5 $\mu g\ m^{-3}$, respectively. The annual mean $NO_2$ and $PM_{2.5}$ concentrations across the network exceed the updated WHO guidelines by 7 $\mu g\ m^{-3}$ for $NO_2$ and 3 $\mu g\ m^{-3}$ for $PM_{2.5}$. However, none of the sites had values exceeding the legally binding UK/EU standards. An excess concentration of 12.5 $\mu g\ m^{-3}$ of $NO_2$ in the network was seen relative to background levels measured by the regional monitoring station at Ladybower reservoir. This highlights the risk of pollution exposure for individuals due to local sources and supports the use of local monitoring to characterize the risk.

Spectral analysis is found to be a good method for studying the variation within the time series. This approach enabled the detection of different underlying periodicities in time series data and allowed the pollution signal to be apportioned to different categories of pollution source whether local, urban or regional. The results highlighted the advantages of having a densely deployed sensor network over the sparse conventional air quality monitoring stations. The highly increased spatio-temporal resolution of low cost sensors combined with their dense placement near pollution sources makes it possible to provide additional information on the patterns and sources of air pollution, which in turn provides a better description of the highly variable and complex nature of pollution.

*Data availability.* All raw data is available upon request



*Author contributions.* Conceptualization: MSJ, JAS. Methodology: LBF, MSJ, JAS. Software: JAS, RS. Validation: LBF, RS, JAS. Formal analysis: LBF. Investigation: LBF. Resources: MSJ. Data curation: LBF. Writing–original draft preparation: LBF. Writing–review and editing: LBF, MSJ, RS, JAS, OH. Visualization: LBF. Supervision: JAS, OH, MSJ. Project administration: MSJ, JAS. Funding acquisition: MSJ, OH.

*Competing interests.* RS, JAS and MSJ are employees of AirLabs and LBF is partly funded by AirLabs.

*Acknowledgements.* The authors thank Bartosz Gaik and Archie Waller for their dedicated and professional assistance with logistical and technical issues. LBF is supported by BERTHA - the Danish Big Data Centre for Environment and Health funded by the Novo Nordisk Foundation Challenge Programme (grant NNF17OC0027864). The authors acknowledge ©OpenStreetMap contributors 2021 for map data, available from https://www.openstreetmap.org, and David Carslaw for providing the online available openair package in R.



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




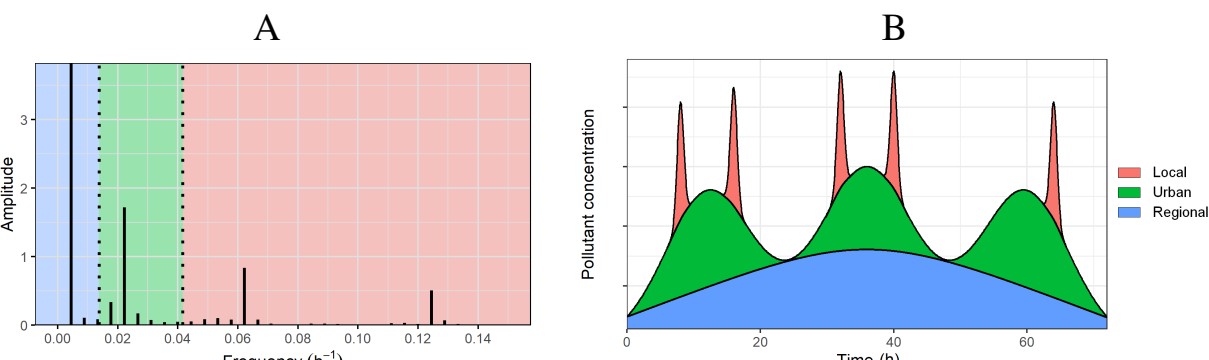

**Figure 2. A.** A periodogram showing short-term fluctuations at the high frequencies (red), background signals at low frequencies (blue) and the fluctuations in-between (green). **B.** Schematic illustration of air pollutant contribution from regional transport (blue), the urban area (green), and the street (red). The relative concentration of the contributions depends on the considered pollutant and the dispersion conditions.

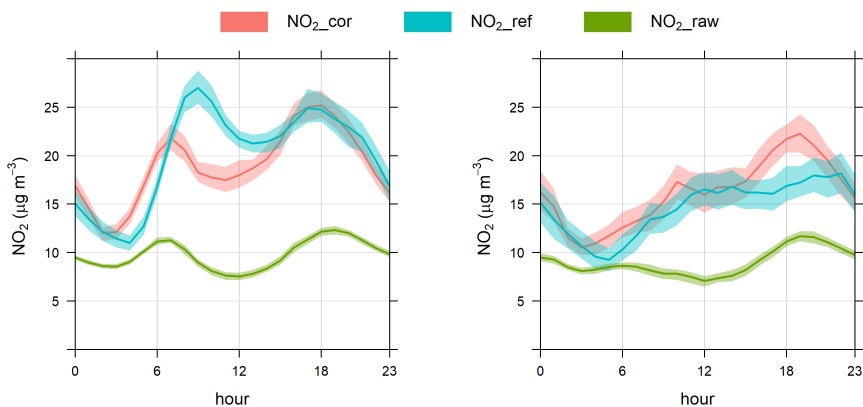

**Figure 3.** Daily patterns of $NO_2$ measured by the reference instrument (blue), corrected $NO_2$ concentrations (red) measured by AirNode4P01 and uncorrected $NO_2$ concentrations (green) measured by AirNode4P01 during weekdays (left) and weekends (right). Note the different scale y-axis. The shading shows the 95% confidence intervals of the mean. The plots are produced by `timeVariation{openair}` (Carslaw, 2012).



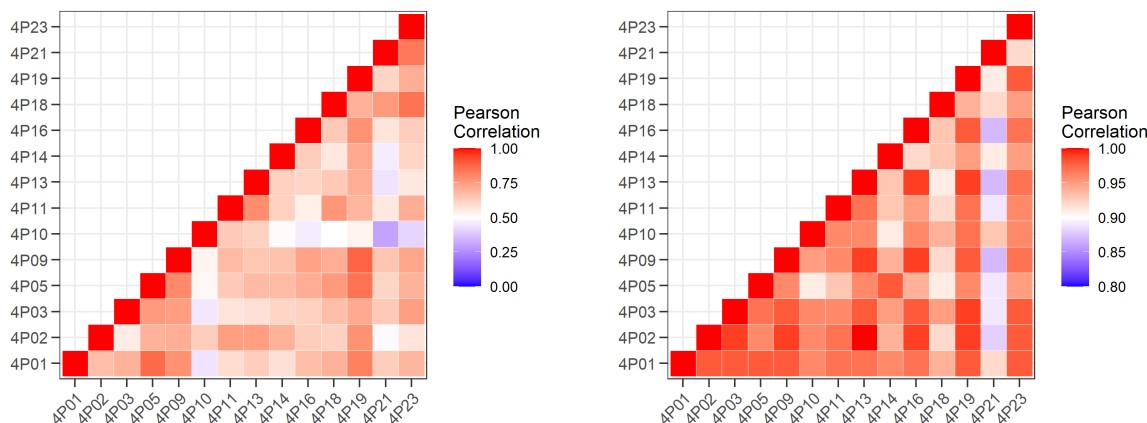

**Figure 4.** Heatmap of the Pearson correlation coefficient matrix for $NO_2$ (left) and $PM_{2.5}$ (right). Note the different scales.

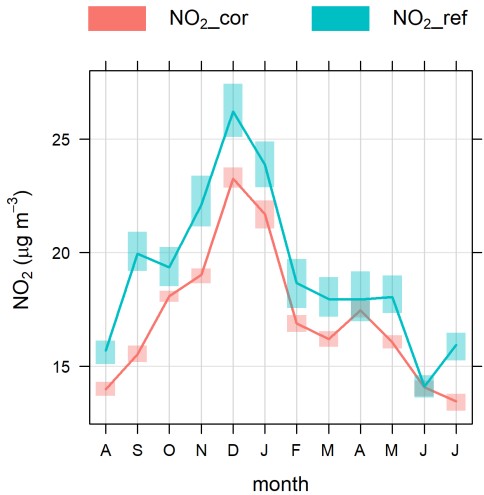

**Figure 5.** Monthly variation of $NO_2$ measured by AirNode4P10 (red) and the corresponding reference instrument at Stoke-on-Trent Center (blue). The plot is produced by `timeVariation{openair}` (Carslaw, 2012).



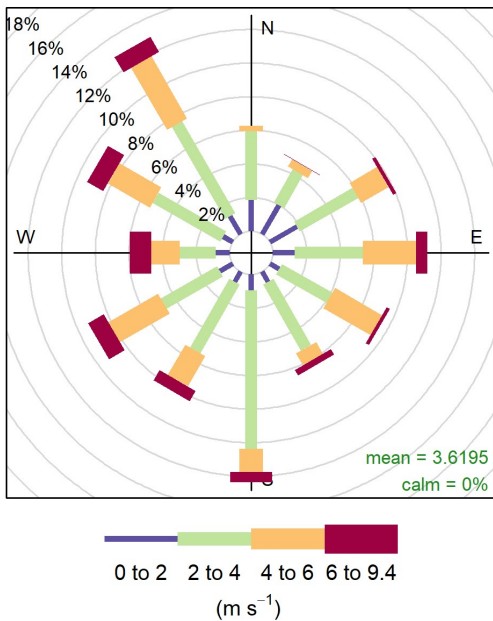

**Figure 6.** Windrose showing the frequency of counts by wind direction (%). The plot is produced by `windRose{openair}` (Carslaw, 2012).

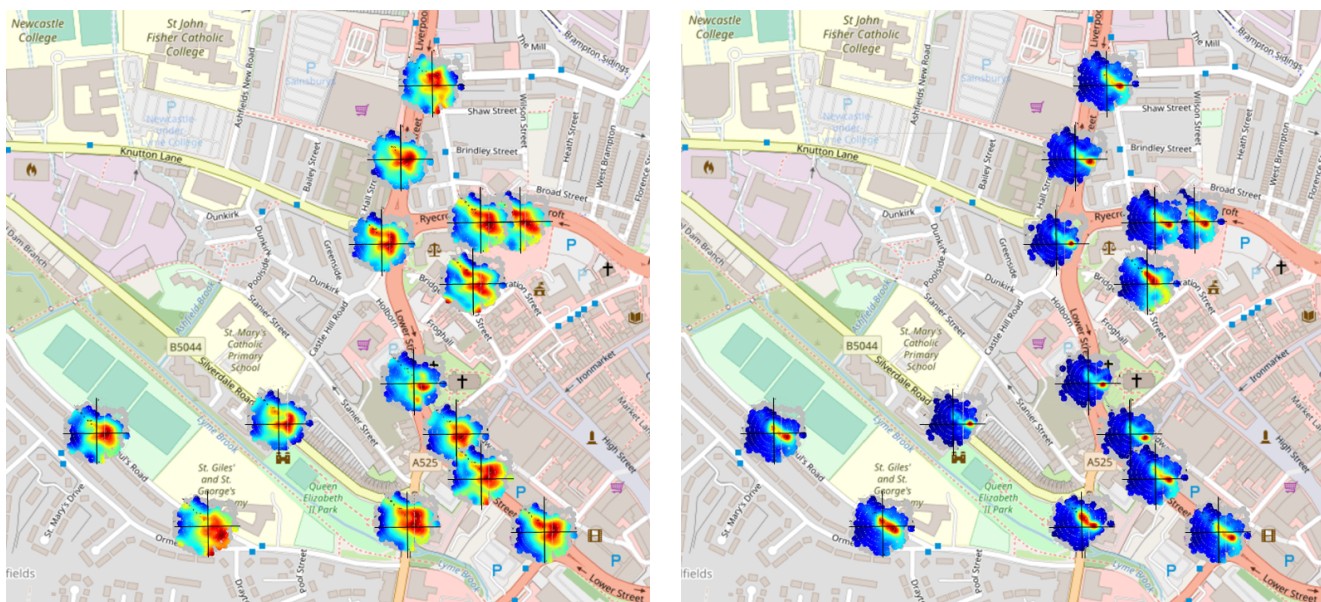

**Figure 7.** Bivariate polar plots of NO$_2$ (left) and PM$_{2.5}$ (right) show the spatial variability in the study area for the entire study period. The figures are produced by `polarMap{openairmaps}` (`openairmaps` is a package that supports `openair` (Carslaw, 2012) for plotting on various maps), where the maps are obtained from ©OpenStreetMap contributors 2021. Distributed under the Open Data Commons Open Database License (ODbL) v1.0 (OpenStreetMap, 2021).





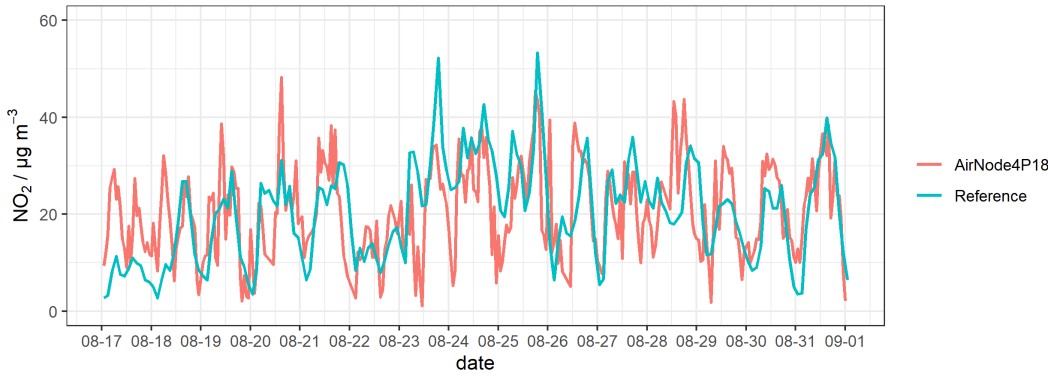

**Figure 8.** Time series of NO$_2$ measured by AirNode4P18 with a time-resolution of 30 minutes and the corresponding reference instrument at Stoke-on-Trent Center with a time-resolution of 1 hour. For clarity, only two weeks of data are shown.

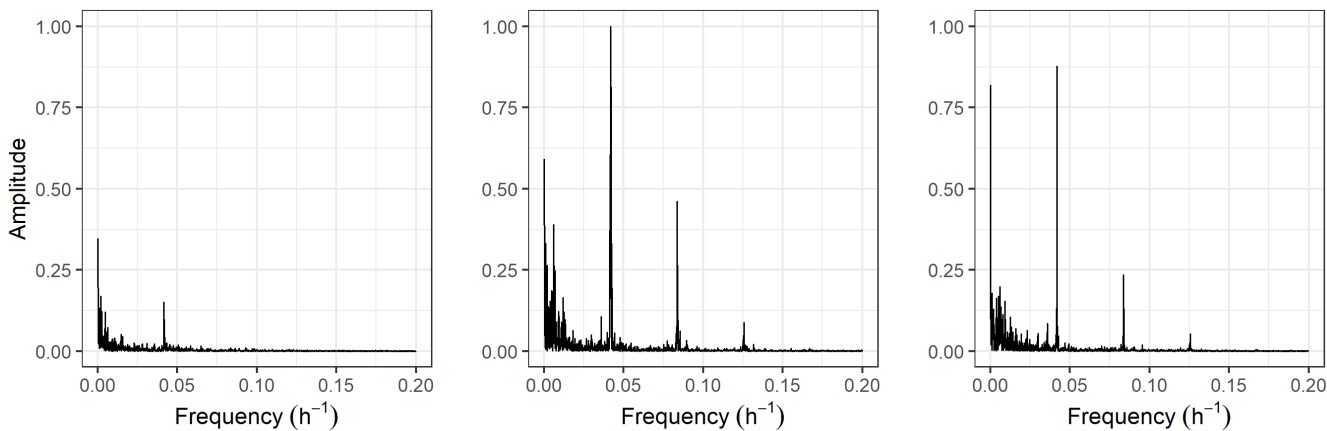

**Figure 9.** Periodograms for NO$_2$ at regional background AQMS, Ladybower (left), street AQMS, Stoke-on-Trent A50 Roadside (middle) and urban background AQMS, Stoke-on-Trent Centre (right). All periodograms are normalized against the highest peak.



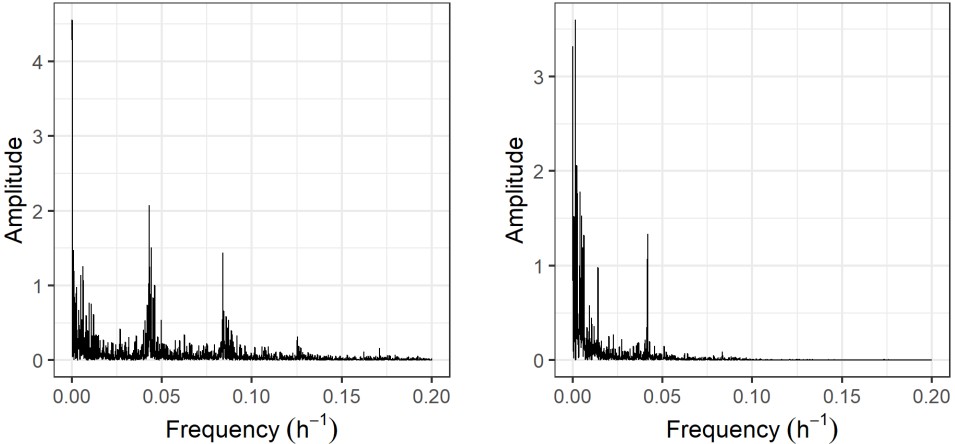

**Figure 10.** Periodogram of $NO_2$ (left) and $PM_{2.5}$ (right) obtained by AirNode4P01.



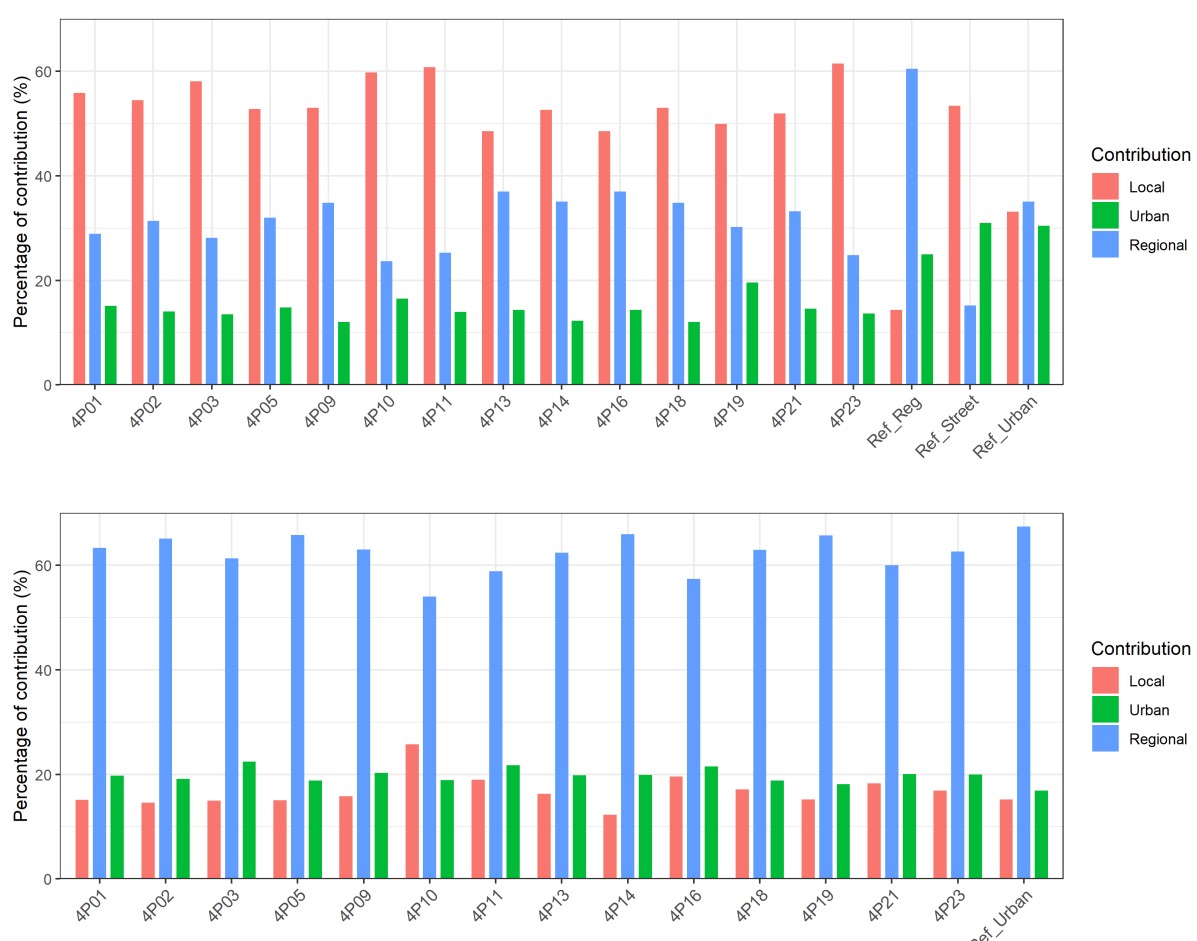

**Figure 11.** Histogram of percentages of contribution (%) of local (red), urban (green) and regional sources (blue) for $NO_2$ (top) and $PM_{2.5}$ (bottom) measured by all AirNodes in the network as well as for the three AQMS. *Ref_Reg* is the regional background AQMS, Ladybower, *Ref_Street* is the street AQMS, Stoke-on-Trent A50 Roadside, and *Ref_Urban* is the urban background AQMS, Stoke-on-Trent Centre.