# Peer review of "Are dense networks of low-cost nodes really useful for monitoring air pollution? A case study in Staffordshire"

_EGUsphere, 2022_

## Author Response (AR2)

Thank you for the review and please find our replies below.

Reviewer 1:

4. **Are the scientific methods and assumptions valid and clearly outlined?**

   Yes., but, some suggestions for clarifying some method descriptions given below + requirement: a clear statement on the fact, that the current study does not try to scientifically evaluate the presented methods (yet) + it does not prove that the sensors used pass any concrete quality criteria . These statements should be added/corrected

Reply:

Thank you for the suggestion, we have added two passages to the manuscript in the Introduction.

'This paper does not attempt to demonstrate that the low-cost air pollution sensors meet specific air quality monitoring standards. Rather, we argue that data obtained from such a network is able to provide useful additional information about local air pollution that extends what can be learned from conventional air quality monitoring stations.'

and

'The approach of frequency domain analysis will be further evaluated in subsequent studies.'

In addition we have added this passage to the Conclusions

'Therefore, this study, like many others, clearly indicates that while low-cost air pollution sensors can be useful, calibration and correction is far from trivial and requires supporting data from reference stations. The corrected $NO_2$ concentrations have a strong connection with the reference station used so results reflect both the low-cost air pollution sensor data and reference station data.'

Reviewer 1:

7. **Does the title clearly reflect the contents of the paper?**

   I would probably change the wording slightly .. e.g." better at"-> " really useful for"/ but this maybe just matter of taste ?

Reply:

Thank you, we like the suggestion and have changed the title accordingly.

Reviewer 1:

**ADDITIONAL NOTES /COMMENTS/SUGGESTIONS:**

All in all, the paper presents results and partly innovative analyses from a very interesting measurement campaign, and as such is a valuable contribution supporting for a further work related to understanding, interpreting and calibrating sensor network results in the future.

Some specific notes/suggestions, which would make the paper even stronger.

1. It should be clearly stated if **ALL** data from the reference station is used for calculating the temperature, scale and offset correction; or is some of the data left for evaluation?

   Reply: Thank you for the comment. All data from the reference is used for the correction, because it gave the best correction and a strong evaluation would require a different data set.

2. Is there any reference or justification for the choice of the specific quantiles (20, 80, 25) used for scale & offset correction?

   Reply: The justification is based on consideration of a Gaussian distribution of the concentration profiles, where the local sources will contribute to the smaller quantiles and long-transport/regional sources will contribute to the quantiles in the middle of the distribution. Different quantiles were tested, and we found that this combination gave the best correction. Also, we now write, 'The local-cutoff is chosen based on the European Environment Agency's definition of local time scale \citep{EEA2008}.' And 'The cutoff frequency for the regional contribution is based on the intercontinental transport, which occurs on timescales on the order of three days to one month \citep{Stohl2002}.'

3. As the temperature correction /regression fit is obviously quite crucial for the whole study, it would be very important to see at least some summary statistics related to the "goodness" of the fit: naturally, if all the available data was already used for correcting the concentrations no real evaluation is possible, but e.g. adding the reference station data/correlations to figure 4 in addition to the sensors would at least give some rough idea how well the temperature correction works

   Reply: Thank you for the comment. All available data was used for the correction, so as you say, no independent evaluation can be performed. As you suggest, the correlation with the reference readings (4 km away) are added to the heat correlation matrices, and it is seen how the AirNode $NO_2$ readings slightly correlate with the reference data. We would not expect much higher correlation due to the relatively large distance between the AirNodes and the reference.

4. The correction coefficients: some short explanation/discussion e.g. on the value of a3 (~1000).. This seems to mean that the sensor raw-"concentration" is given in completely different units? (which should reflect on fig 3 ?)

   Reply: Thank you for the comment. It has been written more clearly now. The raw sensor output is in voltage and what you see in Figure 3 is a laboratory-calibrated value. This was done to make the Figure clearer since both $NO_2$_cor, $NO_2$_ref and $NO_2$_raw could be one the same axis.

 + a short comment also on the meaning of a0 value, which is higher than the average concentration? ,

Reply: It is mentioned that it is a property of the Alphasense cell and can vary a lot from cell to cell, which generally highlights that low-cost sensors need to be calibrated or corrected to obtain absolute concentrations. We write, 'The correction coefficient, $a_0$, or offset of the sensor, is higher than the average concentration and it has a relatively high standard deviation. This is a property of the Alphasense cell and can vary significantly from cell to cell, which underscores the importance of calibrating and correcting the raw data from low-cost sensors in order to obtain accurate concentrations.'

+ some "order of magnitude" estimate on the importance of the 4 terms of the equation  - just to get an idea what is important and what is not

Reply: Thank you for the comment. It is mentioned that the p-value for all terms for all AirNodes was below 0.05 meaning they were all important for the $NO_2$ corrections.

5.  This statement should be removed (CH 3.1.3) " In addition, this indicates that the AirNodes meet the specifications of the Class 1 device standard specifying quality objectives for indicative measurements (AQ, 2021). Class 1 dictates that measurement uncertainty should be below 25% for NO2 and 50% for PM2.5.  "

    **.. or** very concrete/new data supporting this should be represented:  currently there is no evidence on this and   the real evidence for this can most probably be achieved only using co-located reference sensors.

    Reply: It is a good point. The paragraph has been removed.

6.  (minor) did not exactly like too much this part.." Using the Einstein-Smoluchowski relation K = d^2/(2t) (Einstein, 1905; Smoluchowski, 1906), we can solve for the   characteristic distance as a function of time, At wind speed of 5 m s-1, after a day, a spike of pollution will take a minimum of 15 minutes to pass.

    My concrete suggestion: this could be safely removed, without losing too much:

to start with: referring to "Einstein et al." ,would also require explaining how this is related to dispersion of pollutants (the analogy may not be so obvious to everyone)  ..and even after that the final statement is just trivial, "plumes will broaden with time..":. putting any concrete numbers in that statement (which seems to be the goal), is dangerous as everything is very much dependent on met-conditions/terrain/local environment, so any values "guessed" for K maybe order-of magnitude wrong, and for this study these numerical values are not too relevant anyway.

    Reply: True, it is a loose, handwaving explanation. It has been removed.

7.  the spectral analyses part was very interesting; actually it was so interesting that it would be very useful to give a short, concrete description of the exact method of "integrating the peaks"  (as a set  of equations, with exact integration limits,  or something similar )

    Reply: Thank you for the suggestion. The integrals have been added to the methodology section

8.  in the conclusions it should be stressed, that this study (like many others) indicates clearly that (LC) sensors CAN be useful, but the calibration/correction of the results is far from trivial and requires  supporting data and reference level station(s)

    Reply: It is added to the conclusion section. 'Therefore, this study, like many others, clearly indicates that while low-cost air pollution sensors can be useful, calibration and correction is far from trivial and requires supporting data from reference stations. The corrected NO$_2$ concentrations have a strong connection with the reference station used so results reflect both the low-cost air pollution sensor data and reference station data.'

9.  in the conclusions: more clear statement on the fact, that the corrected NO2 concentrations have a very strong connection to the one reference stations used for "calibration" .. so whatever is

seen/analyzed from the corrected data, reflects **not only the sensors**, but also strongly the reference station data

Reply: Very nice point. We have added the passage, 'Therefore, this study, like many others, clearly indicates that while low-cost air pollution sensors can be useful, calibration and correction is far from trivial and requires supporting data from reference stations. The corrected $NO_2$ concentrations have a strong connection with the reference station used so results reflect both the low-cost air pollution sensor data and reference station data.'

10. ( very minor ) eq. 9.. also, vk should be explained

   Reply: Done

 Reviewer 2:

**Thank you very much for your comments and please find our replies below.**

**General Comments**

Frederickson et al., presented results of a one-year measurement campaign with 18 low-cost sensors measuring NO2 and PM2.5 measurements in Newcastle-under-Lyme, Staffordshire, UK. They describe a remote calibration strategy for electrochemical NO2 measurements that accounts for the temperature-dependent response of the sensors used. They use spectral analysis to identify different frequencies in the PM and NO2 time series and allocate PM2.5 and NO2 to local, urban, and regional sources based on 3-defined periodicity ranges. In all, this paper highlights a successful measurement campaign and an insightful method for source attribution. This manuscript may be accepted for publication after addressing the following comments:

**Specific Comments**

·   This paper does not claim to validate the results, but a brief comparison to modeled NO2 and PM2.5 concentrations would significantly strengthen the findings and the claim that low-cost sensor networks offer additional benefits and insights beyond the ability of AQMs or expensive sensors. This could also be addressed by simply comparing the results of the Fourier Transform with an emission inventory for the area. Does this result tell us something new about the sources of NO2 and PM2.5 or does it validate the inventory?

**Reply:** Thank you for the comment. The main focus of the paper is to demonstrate that additional useful information can be obtained from low cost sensors, and we have validated the analysis to the extent possible. Unfortunately there is not a detailed emissions inventory for the study location. We have added the following in the Introduction to address the larger question:' This paper does not attempt to demonstrate that the low-cost air pollution sensors meet specific air quality monitoring standards. Rather, we argue that data obtained from such a network is able to provide useful additional information about local air pollution that extends what can be learned from conventional air quality monitoring stations. The data obtained from the low-cost air pollution sensor network is used for time series analysis in the frequency domain to obtain information on the variability of air pollution concentrations and to distinguish local sources from regional. The network, together with the analysis approach, has allowed pollutant emissions

attributable solely to the local sources to be distinguished from other regional or long-range transport sources. The approach of frequency domain analysis will be further evaluated in subsequent studies.'

· Please comment on the remote NO2 sensitivity correction using the monitoring station at Stoke -on-Trent Centre, why do we expect the same variation at this reference site as in Newcastle-under-Lyme? Particularly given the difference in source apportionment between the AirNodes and this reference site seen in Figure 11, is this a reasonable assumption?

**Reply:** Even though they do not have the same variation (which can be seen in Figures 3 and 11), the same correction is applied for all AirNodes, meaning their relative measurements will be useful for the Fourier transform. We have made use of all nearby monitoring stations and clearly, we do not expect, nor do we see, the same local variation due to the distance between the sites. Nonetheless valuable insight is obtained from the comparison.

· No correction used for SDS-011. Can you please comment on the validation of these sensors?

**Reply:** A paragraph has been added to the correction section, where it is explained why no correction is performed on the SDS-011 readings. However, the correlation between $PM_{2.5}$ measured by the AirNodes and the $PM_{2.5}$ measured by the reference are added in the correlation heatmap. 'Regarding the SDS-011 $PM_{2.5}$ readings, outliers were removed by removing all values exceeding 5 times the standard deviation. Scale and offset correction was performed for $PM_{2.5}$ similar to the one for the $NO_2$ readings. However, there was no significant difference between the corrected and uncorrected $PM_{2.5}$ readings since the $PM_{2.5}$ readings were already highly correlated (mean $R^2 = 0.72$) with the reference readings from the Stoke-on-Trent Centre.'

· Please comment on the choice of <1 day and >3 days as the cutoffs for the regional or urban contribution frequencies.

**Reply:** This is added to the manuscript: "The local-cutoff is chosen based on the European Environment Agency's definition of local time scale.
…
The cutoff frequency for the regional contribution is based on the intercontinental transport, which occurs on timescales on the order of three days to one month".

· Line 255. Is the observed difference as expected? Is there a reason to expect the reference to peak 2 hours later in the morning?

**Reply:** It is hard to say exactly why the reference peaks later than what the AirNodes recorded. As we write, the concentration of $NO_2$ can have different profiles at different locations, depending on the traffic modes and sources.

· Figure 3 and 5 show comparison of a single AirNode, are these data characteristic of all of the AirNodes?

**Reply:** Yes, and it is added to the manuscript now. 'All AirNodes have the same tendencies, so \textbf{Figure \ref{fig:Corrected_data}} is characteristic of all AirNodes.'

·    Line 302 is misleading because the seasonal effect is likely still the dominant effect. Which months were impacted by lockdown strategies?

**Reply:** We can see how it is misleading and have decided to delete the passage about Covid-19. It did not contribute to the story.

·    Line 245-252 Can you show more details on the performance of modeled temperature data? On "the correction methodology even with the modeled temperature data, yields corrected readings that follow expected trends, giving confidence in sensor accuracy."  If some sensors are shaded and others are in full sun, the temperature inside the sensor package can vary dramatically from the outside temperature.

**Reply:** Thank you for the comment. It is an important concern. The correlation with the reference readings (4 km away) are added to the heat correlation matrices, and it is seen how the AirNode $NO_2$ readings slightly correlate with the reference data. It is true that the correction would have been more efficient if internal AirNode temperature had been measured.

**Technical Corrections**

Line 175 missing a space: "Q0.25,AirNode"

Line 219 "a upper" -> "an upper"

Line 221 "on Figure 2" -> "in Figure 2"

Line 280 remove comma

Line 334 "speed" -> "speeds"

Line 380 "odccurs" -> "occurs"

**Reply:** Thanks. They are fixed.